# The Impact of Multiple Sclerosis on Work Productivity: A Preliminary Look at the North American Registry for Care and Research in Multiple Sclerosis

**DOI:** 10.3390/neurosci6030082

**Published:** 2025-08-22

**Authors:** Ahya Ali, Kottil Rammohan, June Halper, Terrie Livingston, Sara McCurdy Murphy, Lisa Patton, Jesse Wilkerson, Yang Mao-Draayer

**Affiliations:** 1Department of Neurology, Autoimmunity Center of Excellence, University of Michigan Medical School, Ann Arbor, MI 48109, USA; ahya.ali@wmchealth.org; 2Department of Neurology, Division of Multiple Sclerosis, University of Miami Miller School of Medicine, Miami, FL 33136, USA; krammohan@med.miami.edu; 3Consortium of Multiple Sclerosis Centers (CMSC), Hackensack, NJ 07601, USA; 4Octave Bioscience, Inc., Menlo Park, CA 94025, USA; tlivingston@octavebio.com; 5Social & Scientific Systems, DLH Holdings Corp. Company, Silver Springs, MD 20910, USA; sara.mccurdy@dlhcorp.com (S.M.M.); lisa.patton@dlhcorp.com (L.P.); jesse.wilkerson@dlhcorp.com (J.W.); 6Autoimmunity Center of Excellence, Multiple Sclerosis Center of Excellence, Arthritis and Clinical Immunology Research Program, Oklahoma Medical Research Foundation Oklahoma City, Oklahoma City, OK 73104, USA

**Keywords:** multiple sclerosis, health economics outcomes research, health resource utilization, fatigue, work productivity, financial burden

## Abstract

Objective: We aimed to quantify multiple sclerosis (MS)-related work productivity and to illustrate the longitudinal trends for relapses, disease progression, and utilization of health care resources in a nationally representative cohort of working North Americans living with MS. Background: The North American Registry for Care and Research in Multiple Sclerosis (NARCRMS) is a multicentered physician-reported registry which prospectively collects clinical information including imaging data over a long period of time from people with MS from sites across the U.S. and Canada. The Health Economics Outcomes Research (HEOR) Advisory Group has also incorporated Health-Related Productivity and Health Resource Utilization questionnaires, which collect information about health care economics of people with MS and its effects on daily life. Design/Methods: This is a prospective observational study utilizing data from NARCRMS. Socio-demographic, clinical, and health economic outcome data were collected through previously validated and structured questionnaires. Logistic regression was used to calculate the relative odds of symptom impact, with a generalized logit link for number of relapses. Cox proportional hazards regression was used to calculate hazard ratios for time to first relapse. Results: Six hundred and eighty-two (682) people with MS were enrolled in NARCRMS and had completed the HEOR questionnaires at the time of the analysis. Among the participants, 61% were employed full-time and 11% were employed part time. Fatigue was the leading symptom reported to impact both work and household chores. Among the employed participants, 13% reported having missed work with a median of 6.8 (IQR: 3.0–9.0) missed hours due to MS symptoms (absenteeism), while 35% reported MS having impacted their work output (presenteeism). The odds of higher disease severity (EDSS 2.0–6.5 vs. 0.0–1.5) were 2.29 (95% CI = 1.08, 4.88; *p* = 0.011) times higher for participants who identified reduction of work output. Fatigue was the most identified symptom attributed to work output reduction. Among all participants, 33% reported having missed planned household work with a median of 3.0 (IQR: 2.0–5.0) hours. The odds of higher disease severity were 2.49 (95% CI = 1.37, 4.53; *p* = 0.006) times higher for participants who identified reduction in household work output, and 1.70 (CI = 1.27, 2.49; *p* = 0.006) times higher for those whose fatigue affected housework output as compared to other symptoms. Conclusions: A preliminary review of the first 682 patients showed that people with MS had reduced work and housework productivity even at an early disease state. Multiple sclerosis (MS) can significantly impair individuals’ ability to function fully at work and at home, with fatigue overwhelmingly identified as the primary contributing factor. The economic value of finding an effective treatment for MS-related fatigue is substantial, underscoring the importance of these findings for policy development, priority setting, and the strategic allocation of healthcare resources for this chronic and disabling condition.

## 1. Introduction

Multiple sclerosis (MS) is the leading cause of non-traumatic neurological injury, with a prevalence of 112 per 100,000 people in the US [1]. It is a chronic immune mediated disease of the central nervous system with a high impact on the health-related quality of life of patients (HRQoL). MS has a highly heterogenous disease trajectory, with potential effects on mobility, coordination, cognition, vision, sleep, bowel and bladder dysfunction, and other functional domains [2,3]. All these symptoms often lead to restricted physical activity, reduced work productivity, increased health care resource utilization, and high rates of unemployment among people with MS [4]. Decreased work productivity has also been associated with increased comorbidity rates, and individuals may need more resources and assistance to accomplish daily activities and goals [5].

Since MS is prevalent among the working-age group (20–60 years) during their most productive years, the long-term management of MS has significant direct and indirect financial consequences [6]. Having a diagnosis of MS is approximately 7.5 times more costly than having no chronic condition [7]. The total annual all-cause healthcare costs for MS, as reported in studies, is estimated at $47,215 per patient per year. Direct economic costs include MS-related resource utilization for prescription drugs, inpatient and ambulatory care, non-medical treatments, and medical aids and have been reported in literature to be between $16,000–$34,000 per patient per year [8]. Indirect economic costs are incurred in the form of net financial loss for the patients due to early retirement, disablement, unemployment, or sick leave. A systematic review reported that the annual indirect costs of MS per patient can range from approximately US$2000—due to factors such as sick leave and disability—to as much as US$20,000, primarily driven by underemployment and unemployment related to disability. Early retirement is a major contributor to the overall financial burden of the disease [9].

The economic burden of MS has largely been studied for healthcare resource utilization, work productivity, health-related quality of life, and lifetime cost [10]. The largest cross-sectional study was conducted by the European Multiple Sclerosis Platform [11]. These observational studies had enrollment from patient organizations, and collected patient reported data from surveys [12,13,14,15,16,17,18]. These studies have reported increased costs, with escalating levels of disability [19]. The annual cost of MS care increases substantially with disability severity, as measured by the Kurtzke Expanded Disability Status Scale (EDSS). Reported costs were $5353 per patient for an EDSS score of 0.0–2.5; $11,110 for EDSS 3.0–5.5; $27,807 for EDSS 6.0–7.5; and $49,823 for EDSS 8.0–9.5. The estimated cost at the highest disability level is nearly ten times greater than that at the lowest level [20].

Relapsing-remitting multiple sclerosis (RRMS) is the most common form of MS and accounts for approximately 85% of MS diagnosis. Patients commonly experience acute neurological attacks or exacerbations, i.e., relapses. The estimation annual cost of relapse is $4449 per patient per year. The association of the cost of relapse with the EDSS score has been estimated to be three times more for an EDSS score of 3.0–5.5 as compared to an EDSS score of 0.0–2.5. Disease modifying therapies (DMTs) have been found to reduce relapses and slow disability progression. Previous analysis has shown to reduce relapse-associated costs and improved work productivity outcomes with higher efficacy DMTs compared with lower-efficacy DMTs, with Ocrelizumab demonstrating best outcomes [21]. DMTs have also been found to slow the conversion from clinically isolated syndrome (CIS) to RRMS and secondary progressive MS (SPMS)as progressive disease phenotypes have been associated with higher overall costs [22]. Therefore, the choice of DMT is not only pertinent for disease management but also for disease costs [23].

Despite significant advances in therapeutics, even with the advent of highly efficacious DMT’s, MS continues to be associated with significant long-term neurological functional impairment and disability. In addition to DMTs, 77.6% of patients also used other prescription drugs, predominantly antidepressants (52.7%) and anti-spasticity drugs (48.5%), followed by anti-fatigue medication (38.4%), demonstrating the multi-system consequences of this chronic disease. In addition to unemployment, presenteeism (being present at work but not fully functioning) is particularly high among people with MS and has its own costs. Lack of having a fulfilling career has been demonstrated to be associated with anxiety, depression, and lower quality of life [24]. Therefore, determining which of these factors contribute most to this decreased work productivity can highlight the importance of better diagnostics and therapeutics, which would help alleviate the social and economic struggles of these patients.

There is paucity of data on the detailed impact of RRMS on the work productivity of individuals with MS and factors contributing to the economic burden of disease in the US [25,26]. A cross-sectional general health survey administered online reported an association between increased MS disease severity and greater work and activity impairment and health-related quality of life (HRQoL) [27]. Another cross-sectional survey of physician recruited RRMS patients also reported a significant association between level of disability and health care resource utilization (HCRU) [28]. Lastly, a prospective observational cohort study at Partner’s MS Center (CLIMB) examined work productivity and HRQoL in patients [29]. This study reported a high rate of employment (76%). In this single-centered study, CIS was not differentiated from RRMS, and the study sample was limited in geographic location.

The results of these studies suggest the need for additional research to specifically assess the factors leading to labor force absenteeism and work impairment from presenteeism, among people with RRMS in North America.

The North American Registry for Care and Research in Multiple Sclerosis (NARCRMS) is a multicentered platform and at the time of this study had 20 enrolling sites across the US and Canada. Along with clinical and imaging data, NARCRMS prospectively collects information about the health care economics of people with MS and its effects on daily life. NARCRMS incorporated health-related productivity and health resource utilization questionnaires into the registry. These questionnaires are completed at enrollment, and subsequently annually, and for relapse visits for each patient, providing longitudinal follow-up to these questions which is a unique feature of this study. The disability scores, relapse, and disease type are also physician reported in comparison to patient-reported data from the European studies.

The aims of our study were (a) to quantify MS-related work productivity loss in a nationally representative cohort of working North Americans with MS, (b) to identify the common symptoms that lead to decreased work productivity, and (c) to illustrate the trends longitudinally for relapses and follow-up of our patient population. Our study provides current economic data on MS in the US and Canada, that are important for policy development, priority setting, and management of public health. Analyzing the economic impact of MS offers valuable insights for patients, healthcare providers, payers, and society at large, and can inform more effective allocation of resources toward the care of individuals living with this chronic and debilitating condition.

## 2. Methods

The NARCRMS is a multicenter, physician-reported registry that prospectively collects data from individuals with MS across the United States and Canada. In addition to clinical and imaging information, NARCRMS gathers data on the health economics of MS and its impact on patients’ daily lives. Assessing the economic burden of MS through such data provides valuable insights that can guide the allocation of resources toward patient care. Patients with CIS, RRMS or progressive MS who meet the criteria [Age 18 to 65 years; clear date of MS onset or CIS within the past 15 years; must have an EDSS less than 6.5] are recruited. All patients were diagnosed according to the McDonald 2010 criteria and provided informed consent prior to enrollment. MS patients were recruited at 26 NARCRMS sites from December 2016 to May 2020 and comprised 682 participants in North America. We performed prospective and cross-sectional analysis based on the data.

### Data and Analysis

The survey collected detailed information across multiple domains. Demographic data included age, gender, marital status, residence, economic status, education, and employment status. Family history and risk factors covered autoimmune conditions, cardiovascular disease, diabetes, and healthcare utilization (inpatient care, day admissions, rehabilitation, nursing homes, outpatient consultations, diagnostic tests, MS medications, relapse treatments, and other prescription and non-prescription drugs). Additional items addressed the use of equipment and aids, personal investments, community support (nursing visits, home help, transportation), and informal family care (including lost workdays). Clinical data encompassed EDSS score, disease course (Clinically Isolated Syndrome [CIS], Relapsing-Remitting [RR], Primary Progressive [PP], Secondary Progressive [SP]), disease phase (remission, relapse, progression), disease duration, year of symptom onset and diagnosis, and comorbid conditions. All socio-demographic, clinical, and resource utilization data were prospectively collected using a structured questionnaire.

The primary outcome was the work-related productivity loss at work and household work assessed over a 7-day period before the respondent completed the survey. They included absenteeism (work time missed), presenteeism (impairment at work), and work productivity loss (level of work output reduction). The level of work output reduction due to MS symptoms, e.g., fatigue, cognitive impairment, etc., was measured as a range (0–100%), with higher scores indicating greater productivity reduction. EDSS is a measure of disability and was measured by a trained MS neurologist. Scores range on an ordinal scale from 0.0 to 10.0 with higher scores indicating greater disability. Two EDSS score groups were considered for the analysis: mild disease (EDSS: 0.0–1.5) and moderate to severe disease (EDSS: 2.0–6.5). The recall periods for resource use were varied by resource to enhance the precision of the answers. Employment and household work questions had a short recall period of 7 days and resource utilization questions had a recall period of 3 months.

For continuous variables, median and IQR were calculated due to non-normality. *p*-values for categorical variables were calculated using the Chi-Square test. *p*-values for continuous variables were calculated using the Wilcoxon Rank Sum test (for 2 classification groups) and Kruskal–Wallis test (for 3 classification groups). Nonparametric tests were used due to non-normality of the continuous variables. Logistic regression was used to calculate odds ratios for symptom severity. Multinomial logistic regression with a generalized logit link was used for the number of relapses. Estimates were adjusted for age of diagnosis, gender, race/ethnicity, household income, and disease duration. Cox proportional hazards regression was used to calculate hazard ratios for time to first relapse. The event was first relapse and censoring values were date of termination or date of data query, whichever came first. Estimates were adjusted for age of diagnosis, gender, race/ethnicity, household income, and disease duration.

## 3. Results

### 3.1. Demographics

In the cohort, we had a broad sample of 682 patients with MS (Table 1: Patient Demographics). 73% were females and 83% of the cohort was of Caucasian ethnicity, which follows the prevalence patterns of MS in North America. The median age at the time of diagnosis was 33 years for Caucasian and 30 years for non-Caucasian patients (*p* = 0.013) (Table 2a). The cohort comprised 69% participants below the age of 40 years and 95% below the age of 65 years (retirement age) and therefore it was a suitable sample to draw inference about the working age group. Among the participants, 55% had an educational level of college degree or higher at the time of enrollment. 71% of the participants resided in an urban/suburban/small town setting at the time of the survey, while only 11% were from a rural setting.

The household income was <$75,000 for 44%, ≥$75,000 for 48%, and unknown for 7% of the participants (Table 1: Patient Demographics). A slightly higher percentage of the patients with mild disease (58%) had a household income of ≥$75,000, as compared to only 47% of those with moderate to severe disease (*p* = 0.007). There was income disparity according to race, as 56% of Caucasians and 32% of non-Caucasians had a household income of ≥$75,000 (*p* < 0.001) (Table 2a). Place of residence had similar patterns of disparity with 21% of Caucasians residing in an urban setting in comparison to 41% of non-Caucasians (*p* < 0.001).

#### 3.1.1. MS Characteristics and Related Disability

The median overall disease duration for the cohort was 5 years (IQR: 2–8) (Table 1: Disease Characteristics). Median disease duration was 4 years (IQR: 2–7) for the patients with mild disease and 6 (IQR: 3–9) for moderate to severe disease (*p* < 0.001). No statistically significant relationship was observed between age at diagnosis and mild disease—32 years for mild disease as compared to 34 years for moderate to severe disease. The median EDSS at the time of enrollment was 1.5 (IQR: 1.0–2.5) where 52% of the patients had an EDSS of 1.0–1.5 and 45% were between 2.0–6.5, and therefore, our cohort is largely representative of mild-moderate disease severity consistent with enrollment criteria. 80% of the participants with mild disease were females while 70% of those with moderate to severe disease were females (*p* = 0.005) (Table 2b). Only 11% of the participants were receiving disability income at the time of the survey for a median duration of 4 (IQR: 2–6) years continuously at the time of the survey, which is in keeping with the EDSS scores of the cohort [23].

#### 3.1.2. Employment Status

61% of the participants were employed full time, 11% were employed part time, and 22% were not employed at the time of the survey (Table 1: Impact of Disability on Work/Housework). A higher proportion, 86%, of participants with mild disease were employed at the time of survey as compared to 67% of those with moderate to severe disease (*p* < 0.001). 92% of the patients were scheduled to work in the week prior to the survey and 13% reported having missed work due to MS related symptoms during that week. The median hours of work missed due to MS was 6.8 h (IQR: 3.0–9.0). Among the working individuals, 35% of the patients reported MS having impacted their work output in the week prior to the survey. MS impacted work output for 28% of those with mild disease and 45% of those with moderate to severe disease (*p* < 0.001). Amongst the symptom categories of fatigue, cognition, weakness, pain, and bowel/bladder issues, the greatest impact was identified as fatigue (22%). The odds of higher disease severity (EDSS 2.0–6.5 vs. 0.0–1.5) were 2.29 (95% CI = 1.08, 4.88; *p* = 0.011) times higher for participants who identified reduction of work output.

### 3.2. MS Impact on Housework

86% of the patients reported having planned doing household chores in the week prior to the survey (Table 1: Impact of Disability on Work/Housework). Among them, 33% reported having missed housework due to MS, with a median of 3.0 h (IQR: 2.0–5.0). Among patients with mild disease, 37% reported MS impacting housework output in comparison to 56% of those with moderate to severe disease (*p* < 0.001). There was race disparity, as 31% of Caucasians reported MS impacting housework output as compared to 45% of non-Caucasians (*p* = 0.030) (Table 2a). Amongst the symptom categories, fatigue was the most common symptom (35%) (Table 1: Impact of Disability on Work/Housework) affecting housework output. Fatigue was affected by disease severity, as 32% of those with mild disease reported fatigue as the factor which impacted their housework productivity compared to 45% of those with moderate to severe disease (*p* = 0.002) (Table 2b). The odds of higher disease severity were 2.49 (95% CI = 1.37, 4.53; *p* = 0.006) times higher for participants who identified reduction in household work output (Table 3). The odds of higher disease severity were 1.70 (CI = 1.17, 2.49; *p* = 0.006) times higher for those whose fatigue affected housework output as compared to other symptoms after adjusting for age of diagnosis, gender, race/ethnicity, household income, and disease duration (Table 3).

### 3.3. Indirect Costs of Disability

Only 11% of the patients were receiving disability income continuously with a median of 4 years (IQR: 2–6) (Table 1: Disease Characteristics). Among patients with mild disease, only 4% were receiving disability income in comparison to 19% of those with moderate to severe disease (*p* < 0.001). 18% of the patients needed the help of aids during the three months prior to the survey, and 11% used walking aids. Only 8% of patients with mild disease used aids in comparison with 30% of those with moderate to severe disease (*p* < 0.001). This is reflective of lower disease severity scores of the cohort. A race disparity was seen with 18% of Caucasians utilizing aids as compared to 31% of non-Caucasians (*p* = 0.004) (Table 2a).

### 3.4. Resources Used

Within a 3-month reporting period, 3% of participants had inpatient admissions, while 6% reported day admissions (Table 1: Disease Characteristics). Overall, 81% of participants had at least one consultation with a neurologist during this time. Among paramedical services, physiotherapy was the most commonly utilized. Both resource use and hours were higher among patients with severe disease.

### 3.5. Trends of Number of Relapses

For analysis, the cohort was divided into three categories of 0,1 and 2+ relapses (Table 1: Relapse). 52% of patients had no relapses, 37% had one relapse, and 11% had two or more relapses. A statistically significant relationship was observed between number of relapses and mean age at diagnosis (*p* = 0.035) (Table 2c). The median disease duration was significantly different between those with one relapse (4 years), 1 relapse (3 years) and more than 2 relapses (4.5 years) (*p* = <0.001). Time to first relapse differed according to race and had a median of 12.7 months (IQR: 11.8–14.5) for Caucasians and 14.0 months (IQR: 12.5–16.9) for non-Caucasians respectively (*p* = 0.013) (Table 2a). Fatigue impacted work output more for those with 2 or 3 relapses (28%) as compared to those with one relapse (15%) (*p* = 0.011). Time to first relapse was a median of 10.9 months (IQR 11.9–14.7) and differed according to race with a median of 12.7 months (IQR: 11.8–14.5)

35% of the patients did not change disease modifying agents, 29% had more than one change in therapy and 18% changed therapies more than 4 times, providing us with comparative groups to study the effect of changes of therapy on relapse. The number of DMT changes also had a statistically significant relationship with the number of relapses, where 25% of those with no relapse, 45% of those with 1 relapse, and 57% of those with 2 or more relapses had 2 or more changes in their DMT’s (*p* < 0.001) (Table 2c).

## 4. Discussion

Our study is the first prospective multi-centered study which reports on the MS-related work productivity, loss of work, health care resource utilization, and HRQoL of a North American MS population with mild to moderate MS [30]. Using data from a representative sample MS population, with physician reported disability scores and standardized in person administered surveys, we can elucidate the factors affecting HRQoL that have not been revealed by prior research in this area. This sample includes a higher proportion of participants employed full-time (61%) compared to previous studies, likely reflecting a cohort with lower disability levels and higher educational attainment.

Lower educational attainment, older age, greater disability, longer disease duration, a progressive disease course, and more severe symptoms at onset have all been identified as predictors of unemployment in individuals with MS. This raises the important question of whether the same factors that influence employment status also contribute to reduced work productivity. Little is known about the factors associated with work output loss and reduced productivity in the workplace and at home. Our study, however, highlights an important finding that, even at lower disability levels, people with MS still experience difficulties at work in terms of presenteeism. This information is particularly important when designing interventions to assist people with MS in maintaining or increasing their work productivity and improving their daily quality of life.

In our study, we found that fatigue was the most common symptom (35%) contributing to lower work productivity at work and at home. Problems of fatigue at work have been noted previously in qualitative studies, regional samples and in large databases such as NARCMS and other European cohorts [22,29,31]. However, in these studies there were no clinical measures of disability levels performed, and they were completed online by the patients. The unique strength of our study is the availability of physicians’ assessments of disability in the form of EDSS which provided objective clinical measures of disease severity that were verified against clinician data. We were also able to provide consistent explanations to patients regarding interpreting the wording of questions, as a trained research personnel administered the surveys as opposed to self-interpretation by the patient with online surveys. Another aspect of this cohort is the inclusion of a large sample size with longitudinal data for future studies. The cross-sectional nature of the baseline analysis allows detection of associations between variables at a single point in time. We addressed the likelihood for recall bias by limiting the lookback period to a short one of 7-days. We also assessed factors impacting household work which is a new aspect.

MS fatigue is a complex, multifactorial, and persistent symptom that has been described by patients as experiencing “malaise”, “excessive tiredness”, or “weakness” [31]. However, there is a lack of uniform metrics for evaluating outcome measures of MS fatigue. Since our study elucidated fatigue as the most prominent factor behind loss of work productivity, the economic impact of elucidating the mechanism of MS-related fatigue cannot be overstated [32]. Persistent fatigue has been attributed to irreversible neurodegeneration, whereas fluctuating fatigue is hypothesized to be caused by reversible pathobiological changes (e.g., inflammatory cytokine and hormone levels) [33]. A recent functional imaging study using fluorodeoxyglucose positron emission tomography (FDG-PET) demonstrated reduced glucose metabolism in the bilateral prefrontal cortex and basal ganglia of MS patients experiencing fatigue, compared to those without fatigue [34,35]. Our findings open avenues for further research into the pathological mechanisms whereby fatigue occurs, which could subsequently lead to developments in therapeutics. An important correlation that we discovered was an association between time to relapse and fatigue; further studies may further elucidate the nature of the fatigue, which could potentially change the life of these MS patients.

There are possible limitations to the conclusions that may be drawn. First, the inclusion criteria included only people with MS with 15 years or less of disease duration and EDSS ≤ 6.5 at entry, which was planned to allow for longer term longitudinal follow up. RRMS is the most common type of MS in our cohort, with much smaller numbers of progressive patients. However, better understanding of early CIS and RRMS disease burden may be important for resource allocation. Another limitation is that we did not include a measure of the physical, emotional, and mental demands of the participants’ occupations or their level of interest in their work; since these can be potential confounders, including them in future analysis would be beneficial [36]. Another aspect that was not investigated was the impact of insurance policies and health care coverage on an individual’s employment related decision making. Individuals may limit the number of absentee days to preserve insurance coverage, potentially masking the true extent of work impairment. Future research should explore these dynamics to assess possible interactions. A final limitation of the study is the potential for selection bias, as the analysis was based on a convenience sample. Participants were recruited by neurologists who were managing MS patients in clinic, and the length of the survey may have resulted in the recruitment of a healthier subset of patients in the cohort. However, it is noteworthy that the benefit of physician reported assessments added validity to the clinical data, and more accurate inferences about disease progression could be made.

In summary, this study gives an integrated, contemporary understanding of the economic burden of MS and the complex factors influencing work status changes as well as ability to perform housework. Our findings demonstrate that, despite improvements in workplace accessibility, physical disability and fatigue remain key determinants of changes in employment status among individuals with MS. The results also highlight the importance of EDSS category and relapse frequency in influencing the ability to sustain both paid employment and household responsibilities while managing MS symptoms. These insights are valuable for informing resource allocation and health services planning, particularly as the MS population continues to age. Future research should explore whether earlier identification and treatment of fatigue can enhance productivity at work and home, and support continued participation in the workforce for individuals with MS.

## 5. Conclusions

This study offers a comprehensive, clinician-verified analysis of the impact of multiple sclerosis on work productivity, healthcare resource use, and quality of life among individuals with mild to moderate disease. Drawing on standardized, prospectively collected data from a large North American cohort, we demonstrate that fatigue and physical disability are persistent and significant barriers to sustained employment and effective performance of household tasks, even in those with lower EDSS scores. The association between fatigue, relapse timing, and work impairment highlights critical targets for early intervention. Our findings emphasize the need for improved strategies to assess and manage fatigue in MS and provide a foundation for resource allocation, health policy planning, and development of future longitudinal studies. As the MS population ages, identifying modifiable contributors to productivity loss will be essential for improving both clinical and socioeconomic outcomes.

## Figures and Tables

**Table 1 neurosci-06-00082-t001:** Patient Characteristics. Basic distributions of all variables in HEOR. Continuous variables have median and IQR displayed rather than means due to non-normality.

	*n* (%) or Median (IQR)
**Total**	**682 (100)**
**Patient Demographics**	
Age at Diagnosis (yrs, continuous)	33 (27, 41)
Age at Diagnosis	
12 to 19 years	32 (5)
20 to 29 years	208 (30)
30 to 39 years	229 (34)
40 to 49 years	154 (23)
50 to 59 years	43 (6)
Missing	16 (2)
Gender	
Male	161 (24)
Female	501 (73)
Transgender Male	2 (0)
Missing	18 (3)
Race/ethnicity	
Caucasian	569 (83)
Non-Caucasian	80 (12)
Missing	33 (5)
Educational Attainment	
High School Graduate or Less	159 (24)
Associate’s Degree/Vocational Certificate	141 (21)
Bachelor’s Degree	225 (34)
Master’s/Doctorate/Professional Degree	140 (21)
Household Income	
<$75,000	303 (44)
≥$75,000	330 (48)
Missing	49 (7)
Urbanicity	
Urban	135 (20)
Suburban	172 (25)
Small Town/City	177 (26)
Rural	75 (11)
Missing	123 (18)
Living Situation	
Living with Significant Other & Children	265 (39)
Living with Significant Other	190 (28)
Living with Parent, Sibling, or Other Family	130 (19)
Living Alone	77 (11)
Missing	20 (3)
Changed Insurance in 3 Years Prior	
Yes	222 (33)
No	417 (61)
Missing	43 (6)
**Disease Characteristics**	
Disease Duration (yrs, continuous)	5.0 (2.0, 8.0)
Pain Rating Score	
0	420 (62)
>0	134 (20)
Missing	128 (19)
EDSS at Enrollment (continuous)	1.5 (1.0, 2.5)
EDSS at Enrollment	
0.0 to 1.5	358 (52)
2.0 to 6.5	309 (45)
Missing	15 (2)
Receiving Disability Income	
Yes	73 (11)
No	591 (87)
Missing	18 (3)
Receiving Disability Income (yrs, continuous)	4.0 (2.0, 6.0)
Utilized Aids in 3 Months Prior	
Yes	125 (18)
No	539 (79)
Missing	18 (3)
Type of Aid Utilized in 3 Months Prior	
Walking Aid	75 (11)
Stairlift/Elevator	60 (9)
Modification to Home	23 (3)
Bedlift/Ramp/Rails	22 (3)
Wheelchair	11 (2)
Electric Wheelchair/Scooter	16 (2)
Special Utensils	9 (1)
Modification to Car	4 (1)
Other	7 (1)
Type of Healthcare Provider Visits in 3 Months Prior	
Neurologist	553 (81)
General Practitioner	137 (20)
Ophthalmologist	95 (14)
Physical Therapist	54 (8)
Massage Therapist	53 (8)
Psychiatrist	39 (6)
Psychologist	36 (5)
Occupational Therapist	25 (4)
Chiropractor	19 (3)
Type of Hospitalizations in 3 Months Prior	
Emergency Room Visits	38 (6)
Inpatient Hospitalizations	21 (3)
Rehabilitation Center Admission	4 (1)
Length of Inpatient Hospitalizations (days, continuous)	4.0 (2.0, 7.0)
Number of changes to DMT	
0	237 (35)
1	200 (29)
2	53 (8)
3	71 (10)
4+	121 (18)
**Relapse**	
Time to First Relapse (months, continuous)	12.9 (11.9, 14.7)
Number of Relapses	
0	356 (52)
1	251 (37)
2	72 (11)
3	3 (0)
**Impact of Disability on Work/Housework**	
Employment Status	
Employed Full Time	416 (61)
Employed Part Time	72 (11)
Employed Not Specified	21 (3)
Not Employed	150 (22)
Missing	23 (3)
Scheduled to Work in Week Prior	
Yes	469 (92)
No	38 (7)
Missing	2 (0)
Missed Work due to MS in Week Prior	
Yes	62 (13)
No	403 (86)
Missing	4 (1)
Number of Work Hours Missed (continuous)	6.8 (3.0, 9.0)
MS Impacted Work Output in Week Prior	
Yes	163 (35)
No	301 (64)
Missing	5 (1)
MS Symptom Impacted Work	
Fatigue	101 (22)
Cognition	26 (6)
Weakness	19 (4)
Pain	18 (4)
Bladder/Bowel	3 (1)
Other	48 (10)
Missing	34 (7)
Work Output Reduction	
0%	301 (64)
1–25%	122 (26)
>25%	41 (9)
Missing	5 (1)
Planned Housework in Week Prior	
Yes	589 (86)
No	70 (10)
Missing	23 (3)
Missed Housework due to MS in Week Prior	
Yes	192 (33)
No	394 (67)
Missing	3 (1)
Number of Housework Hours Missed (continuous)	3.0 (2.0, 5.0)
MS Impacted Housework Output in Week Prior	
Yes	264 (45)
No	309 (52)
Missing	16 (3)
MS Symptom Impacted Housework	
Fatigue	209 (35)
Cognition	7 (1)
Weakness	32 (5)
Pain	27 (5)
Bladder/Bowel	4 (1)
Other	44 (7)
Missing	44 (7)
Housework Output Reduction	
0–25%	446 (76)
26–50%	66 (11)
>50%	61 (10)
Missing	16 (3)

**Table 2 neurosci-06-00082-t002:** *p*-values for categorical variables are calculated using the Chi-Square test. *p*-values for continuous variables are calculated using the Wilcoxon Rank Sum test in (**a**,**b**) (where there are 2 classification groups) and using the Kruskal–Wallis test in (**c**) (where there are 3 classification groups). Nonparametric tests are used due to non-normality of the continuous variables. Bold is for statistically significant findings.

(**a**) Bivariate Associations by Race				
	**Caucasian**		**Non-Caucasian**	
	***n*** **(%) or Median (IQR)**		***n*** **(%) or Median (IQR)**	** *p* ** **-Value**
**Total**	**569 (100)**		**80 (100)**	
Age at Diagnosis (yrs, continuous)	**33 (27, 41)**		**30 (25, 37)**	**0.013**
Gender				0.326
Male	141 (25)		16 (20)	
Female	422 (75)		64 (80)	
Educational Attainment				0.779
High School Graduate or Less	136 (24)		17 (22)	
Associate’s Degree/Vocational Certificate	122 (21)		17 (22)	
Bachelor’s Degree	194 (34)		25 (32)	
Master’s/Doctorate/Professional Degree	116 (20)		20 (25)	
Household Income				**<0.001**
<$75,000	**239 (44)**		**50 (68)**	
≥$75,000	**303 (56)**		**24 (32)**	
Urbanicity				**0.001**
Urban	**98 (21)**		**28 (41)**	
Suburban	**151 (32)**		**19 (28)**	
Small Town/City/Rural	**227 (48)**		**21 (31)**	
Changed Insurance in 3 Years Prior				0.530
Yes	188 (34)		29 (38)	
No	357 (66)		47 (62)	
Disease Duration (yrs, continuous)	5.0 (2.0, 8.0)		5.0 (2.0, 9.0)	0.796
Pain Rating Score				0.074
0	107 (23)		22 (33)	
>0	354 (77)		44 (67)	
EDSS at Enrollment (continuous)	1.5 (1.0, 2.0)		2.0 (1.0, 2.5)	0.079
EDSS at Enrollment				0.089
0.0 to 1.5	307 (55)		35 (45)	
2.0 to 6.5	250 (45)		43 (55)	
Receiving Disability Income				0.121
Yes	59 (10)		13 (16)	
No	507 (90)		67 (84)	
Receiving Disability Income (yrs, continuous)	5.0 (2.0, 6.0)		3.0 (2.0, 5.0)	0.345
Utilized Aids in 3 Months Prior				**0.004**
Yes	**99 (18)**		**25 (31)**	
No	**466 (82)**		**55 (69)**	
Number of changes to DMT				**<0.001**
0	**174 (31)**		**44 (55)**	
1	**172 (30)**		**21 (26)**	
2+	**223 (39)**		**15 (19)**	
Time to First Relapse (months, continuous)	**12.7 (11.8, 14.5)**		**14.0 (12.5, 16.9)**	**0.013**
Number of Relapses				0.091
0	287 (50)		42 (53)	
1	223 (39)		24 (30)	
2+	59 (10)		14 (18)	
Employment Status				0.141
Employed	440 (78)		56 (71)	
Not Employed	122 (22)		23 (29)	
Missed Work due to MS in Week Prior				0.681
Yes	56 (14)		6 (12)	
No	348 (86)		45 (88)	
Number of Work Hours Missed (continuous)	6.8 (3.0, 9.5)		6.0 (3.0, 9.0)	0.962
MS Impacted Work Output in Week Prior				0.611
Yes	144 (36)		16 (32)	
No	260 (64)		34 (68)	
Fatigue Impacted Work				0.665
Yes	88 (23)		12 (26)	
No	291 (77)		34 (74)	
Work Output Reduction				0.764
0%	260 (64)		34 (68)	
1–25%	108 (27)		13 (26)	
>25%	36 (9)		3 (6)	
Missed Housework due to MS in Week Prior				**0.030**
Yes	**158 (31)**		**29 (45)**	
No	**349 (69)**		**36 (55)**	
Number of Housework Hours Missed (continuous)	3.0 (2.0, 6.0)		2.0 (2.0, 5.0)	0.374
MS Impacted Housework Output in Week Prior				0.167
Yes	222 (45)		35 (54)	
No	274 (55)		30 (46)	
Fatigue Impacted Housework				0.149
Yes	176 (37)		29 (47)	
No	296 (63)		33 (53)	
Housework Output Reduction				0.872
0–25%	388 (78)		49 (75)	
26–50%	55 (11)		8 (12)	
>50%	53 (11)		8 (12)	
(**b**) Bivariate Associations by EDSS Category				
	**EDSS: 0.0 to 1.5**		**EDSS: 2.0 to 6.5**	
	***n*** **(%) or Median (IQR)**		***n*** **(%) or Median (IQR)**	** *p* ** **-Value**
**Total**	**358 (100)**		**309 (100)**	
Age at Diagnosis (yrs, continuous)	32 (26, 39)		34 (28, 42)	0.247
Gender				**0.005**
Male	**72 (20)**		**89 (30)**	
Female	**280 (80)**		**208 (70)**	
Race/ethnicity				0.089
Caucasian	307 (90)		250 (85)	
Non-Caucasian	35 (10)		43 (15)	
Educational Attainment				**<0.001**
High School Graduate or Less	**61 (17)**		**96 (32)**	
Associate’s Degree/Vocational Certificate	**61 (17)**		**75 (25)**	
Bachelor’s Degree	**145 (41)**		**76 (26)**	
Master’s/Doctorate/Professional Degree	**87 (25)**		**50 (17)**	
Household Income				**0.007**
<$75,000	**141 (42)**		**152 (53)**	
≥$75,000	**192 (58)**		**134 (47)**	
Urbanicity				0.544
Urban	72 (24)		57 (23)	
Suburban	95 (32)		71 (29)	
Small Town/City/Rural	130 (44)		120 (48)	
Changed Insurance in 3 Years Prior				0.753
Yes	118 (35)		103 (36)	
No	221 (65)		183 (64)	
Disease Duration (yrs, continuous)	**4.0 (2.0, 7.0)**		**6.0 (3.0, 9.0)**	**<0.001**
Pain Rating Score				**<0.001**
0	**262 (85)**		**148 (64)**	
>0	**48 (15)**		**83 (36)**	
Receiving Disability Income				**<0.001**
Yes	**13 (4)**		**58 (19)**	
No	**339 (96)**		**240 (81)**	
Receiving Disability Income (yrs, continuous)	3.0 (2.0, 6.0)		4.5 (2.0, 6.0)	0.994
Utilized Aids in 3 Months Prior				**<0.001**
Yes	**28 (8)**		**90 (30)**	
No	**324 (92)**		**208 (70)**	
Number of changes to DMT				0.227
0	131 (37)		95 (31)	
1	99 (28)		100 (32)	
2+	128 (36)		114 (37)	
Time to First Relapse (months, continuous)	12.7 (11.9, 14.7)		12.9 (12.0, 14.6)	0.856
Number of Relapses				0.381
0	191 (53)		152 (49)	
1	125 (35)		124 (40)	
2+	42 (12)		33 (11)	
Employment Status				**<0.001**
Employed	**300 (86)**		**198 (67)**	
Not Employed	**50 (14)**		**97 (33)**	
Missed Work due to MS in Week Prior				0.222
Yes	33 (12)		28 (16)	
No	246 (88)		149 (84)	
Number of Work Hours Missed (continuous)	8.0 (3.0, 12.0)		5.8 (3.0, 9.0)	0.431
MS Impacted Work Output in Week Prior				**<0.001**
Yes	**78 (28)**		**80 (45)**	
No	**201 (72)**		**96 (55)**	
Fatigue Impacted Work				0.398
Yes	56 (22)		42 (25)	
No	203 (78)		125 (75)	
Work Output Reduction				**0.001**
0%	**201 (72)**		**96 (55)**	
1–25%	**57 (20)**		**63 (36)**	
>25%	**21 (8)**		**17 (10)**	
Missed Housework due to MS in Week Prior				**<0.001**
Yes	**79 (25)**		**105 (41)**	
No	**238 (75)**		**154 (59)**	
Number of Housework Hours Missed (continuous)	3.0 (2.0, 5.0)		3.0 (2.0, 6.0)	0.264
MS Impacted Housework Output in Week Prior				**<0.001**
Yes	**113 (37)**		**144 (56)**	
No	**196 (63)**		**111 (44)**	
Fatigue Impacted Housework				**0.002**
Yes	**95 (32)**		**110 (45)**	
No	**198 (68)**		**133 (55)**	
Housework Output Reduction				**<0.001**
0–25%	**262 (85)**		**177 (69)**	
26–50%	**25 (8)**		**39 (15)**	
>50%	**22 (7)**		**39 (15)**	
(**c**) Bivariate Associations by Number of Relapses				
	**0 Relapses**	**1 Relapse**	**2+ Relapses**	
	***n*** **(%) or Median (IQR)**	***n*** **(%) or Median (IQR)**	** *n* ** **(%) or Median (IQR)**	** *p* ** **-Value**
**Total**	**356 (100)**	**251 (100)**	**75 (100)**	
Age at Diagnosis (yrs, continuous)	**33 (27, 41)**	**34 (28, 42)**	**30 (26, 37)**	**0.035**
Gender				0.226
Male	76 (23)	61 (24)	24 (32)	
Female	261 (77)	189 (76)	51 (68)	
Race/ethnicity				0.091
Caucasian	287 (87)	223 (90)	59 (81)	
Non-Caucasian	42 (13)	24 (10)	14 (19)	
Educational Attainment				0.575
High School Graduate or Less	83 (24)	63 (25)	13 (17)	
Associate’s Degree/Vocational Certificate	72 (21)	52 (21)	17 (23)	
Bachelor’s Degree	119 (35)	76 (30)	30 (40)	
Master’s/Doctorate/Professional Degree	66 (19)	59 (24)	15 (20)	
Household Income				0.113
<$75,000	161 (50)	103 (43)	39 (55)	
≥$75,000	161 (50)	137 (57)	32 (45)	
Urbanicity				**0.048**
Urban	**78 (29)**	**42 (19)**	**15 (22)**	
Suburban	**87 (32)**	**67 (30)**	**18 (27)**	
Small Town/City/Rural	**106 (39)**	**112 (51)**	**34 (51)**	
Changed Insurance in 3 Years Prior				0.770
Yes	111 (33)	86 (36)	25 (36)	
No	221 (67)	152 (64)	44 (64)	
Disease Duration (yrs, continuous)	**4.0 (1.0, 8.0)**	**6.0 (3.0, 9.0)**	**4.5 (2.0, 7.0)**	**<0.001**
Pain Rating Score				0.139
0	247 (78)	125 (71)	48 (80)	
>0	70 (22)	52 (29)	12 (20)	
EDSS at Enrollment (continuous)	1.5 (1.0, 2.5)	1.5 (1.0, 2.5)	1.5 (1.0, 2.0)	0.267
EDSS at Enrollment				0.381
0.0 to 1.5	191 (56)	125 (50)	42 (56)	
2.0 to 6.5	152 (44)	124 (50)	33 (44)	
Receiving Disability Income				0.605
Yes	39 (12)	24 (10)	10 (13)	
No	300 (88)	266 (90)	65 (87)	
Receiving Disability Income (yrs, continuous)	4.0 (1.0, 6.0)	5.0 (3.0, 6.0)	2.5 (1.0, 5.0)	0.462
Utilized Aids in 3 Months Prior				0.951
Yes	64 (19)	48 (19)	13 (18)	
No	276 (81)	202 (81)	61 (82)	
Number of changes to DMT				**<0.001**
0	**158 (44)**	**64 (25)**	**15 (20)**	
1	**110 (31)**	**73 (29)**	**17 (23)**	
2+	**88 (25)**	**114 (45)**	**43 (57)**	
Time to First Relapse (months, continuous)	NA	12.8 (12.0. 14.6)	13.2 (11.6, 14.7)	0.805
Employment Status				0.697
Employed	259 (77)	194 (79)	56 (75)	
Not Employed	79 (23)	52 (21)	19 (25)	
Missed Work due to MS in Week Prior				0.097
Yes	39 (17)	17 (9)	6 (12)	
No	195 (83)	162 (91)	46 (88)	
Number of Work Hours Missed (continuous)	8.0 (3.0, 12.0)	4.0 (3.0, 8.0)	4.0 (4.0, 8.0)	0.147
MS Impacted Work Output in Week Prior				0.624
Yes	87 (37)	58 (33)	18 (35)	
No	147 (63)	120 (67)	34 (65)	
Fatigue Impacted Work				**0.011**
Yes	**63 (28)**	**25 (15)**	**13 (28)**	
No	**163 (72)**	**138 (85)**	**33 (72)**	
Work Output Reduction				**0.006**
0%	**147 (63)**	**120 (67)**	**34 (65)**	
1–25%	**55 (24)**	**51 (29)**	**16 (31)**	
>25%	**32 (14)**	**7 (4)**	**2 (4)**	
Missed Housework due to MS in Week Prior				0.219
Yes	107 (36)	64 (29)	21 (33)	
No	192 (64)	160 (71)	42 (67)	
Number of Housework Hours Missed (continuous)	3.0 (2.0, 5.0)	3.0 (2.0, 6.0)	2.0 (2.0, 3.0)	0.239
MS Impacted Housework Output in Week Prior				0.758
Yes	138 (48)	98 (44)	28 (45)	
No	152 (52)	123 (56)	34 (55)	
Fatigue Impacted Housework				0.172
Yes	115 (41)	70 (33)	24 (44)	
No	166 (59)	139 (67)	31 (56)	
Housework Output Reduction				0.348
0–25%	222 (77)	172 (78)	52 (84)	
26–50%	31 (11)	30 (14)	5 (8)	
>50%	37 (13)	19 (9)	5 (8)	

**Table 3 neurosci-06-00082-t003:** Cox proportional hazards regression used to calculate hazard ratios. Event was first relapse and censoring values were date of termination or date of data query, whichever came first. Estimates are adjusted for age of diagnosis, gender, race/ethnicity, household income, and disease duration. Adjusted association of fatigue and inability to work/do housework with disease severity. Bold is for statistically significant findings.

		Disease Severity (REF = EDSS: 0.0 to 1.5)	
		EDSS: 2.0 to 6.5	
Effect	Unit	OR (95% CI)	*p*-Value
Fatigue affects work output	Yes vs. No	1.25 (0.77–2.03)	0.375
Fatigue affects housework output	Yes vs. No	**1.70 (1.17–2.49)**	**0.006**
MS kept from work	1 h	1.02 (0.97–1.08)	0.357
MS kept from housework	1 h	**1.13 (1.04–1.23)**	**0.003**
Work output reduction	>25% vs. 0%	**2.29 (1.08–4.88)**	**0.011**
Housework output reduction	>50% vs. 0–25%	**2.49 (1.37–4.53)**	**0.006**

## Data Availability

We collected and generated the data which are new based on our study.

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
