# Peer review of "The Impact of Multiple Sclerosis on Work Productivity: A Preliminary Look at the North American Registry for Care and Research in Multiple Sclerosis"

_neurosci, 2025, doi:10.3390/neurosci6030082_

Round 1
Reviewer 1 Report
Comments and Suggestions for Authors
Multiple sclerosis is an inflammatory and neurodegenerative neurological disease that affect young patients and has a great impact on their working capabilities and household activities. The economic burden of this disease result not only of direct costs of medication, hospital admittance, but also on indirect cost due to absenteeism at the workplace or presenteeism (with lack of work productivity).
In a prospective, longitudinal study, the authors presented socio-demographic, clinical, health economic data from 682 patients enrolled in The North American Registry for Care and Research in Multiple Sclerosis (NARCRMS). Health-Related Productivity and Health Resource Utilization questionnaires were completed. Logistic regression was used to calculate the relative odds of symptom impact.
The authors found that fatigue was the first symptom that impacted both work and household chores.
Persistent fatigue could be due to irreversible neurodegeneration, but on the other hand fluctuating fatigue is hypothesized to be caused by reversible inflammatory changes.
The study revealed that patients with MS had reduced work and housework productivity even at early disease state.
Author Response
Yes above is a summary of our work. Thank you for your review.
Reviewer 2 Report
Comments and Suggestions for Authors Dear Authors, The article is interesting and generally well-written. However, there are still some major issues that need to be addressed. I recommend including additional bivariate comparisons that are particularly relevant in the context of MS, such as male vs. female, white-collar vs. blue-collar workers, and disease duration. It would also be valuable to present data on the use of disease-modifying therapies (DMTs) and explore whether these are associated with occupational outcomes. Finally, it is crucial that the authors place their findings within a global context. The American population represents only a small portion of the world and does not reflect global diversity. MS is a heterogeneous disease, and occupational outcomes are highly sensitive to country-specific socioeconomic and healthcare factors. I strongly encourage you to address this point in the discussion and cite the following reference: doi: 10.1371/journal.pone.0272156.Author Response
Thankyou so much for your review. The NARCRM database registry includes centers across US and Canada so are more broadly encompassing than a multi-centered American registry. Unfortunately, we do not have details of work description i.e whitecollar versus blue collar captured in the registry, however that would answer a very interesting question. The correlation of use of DMT’s would make for future analysis to correlate with the work productivity data and other clinical information which could be topic of future papers.
Reviewer 3 Report
Comments and Suggestions for Authors
In this article, the authors investigate the role y multiple sclerosis-related work productivity and to illustrate the longitudinal trends for relapses, disease progression, and utilization of health care resources in a nationally representative cohort of working North Americans living with MS. Enrolled 682 people with MS were enrolled in. NARCRMS. The authors conclude that 682 patients showed that people with MS had reduced work and housework productivity even at early disease state and that fatigue is the main problem.
The manuscript is well articulated. The sample is relatively large and representative of the North American population with early/moderate MS. The topic is of great interest.
Despite that, there are some questions.
-No data has been included on the type of work performed, the physical/mental load, and the flexibility or support in the workplace. It is suggested to include this missing data which is of fundamental importance to contextualize the subjects analyzed.
-Fatigue is reported but not measured with validated scales (e.g. MFIS- Modified Fatigue Impact Scale or FSS-Fatigue Severity Scale),limiting accuracy. Please enter this data. Similarly, in the table are reported MS Symptom Impacted Work, , Cognition, Weakness, Pain, but are not reported results. from the scales. Are there data relating to the quantification of pain? of weakness? And relative to cognitive data?
- The percentage of the non-Caucasian population who does it include? enter details.
- Since this is an American study, there are no data on the effect of insurance policies and health coverage on employment and absenteeism. Please clarify this.
- Check the values ​​reported in the text and in the tables.
- Finally, this is a complex work, with long tables, it is suggested to create a summary diagram, with the key points, as a graphical abstract.
Author Response
Comments: In this article, the authors investigate the role y multiple sclerosis-related work productivity and to illustrate the longitudinal trends for relapses, disease progression, and utilization of health care resources in a nationally representative cohort of working North Americans living with MS. Enrolled 682 people with MS were enrolled in. NARCRMS. The authors conclude that 682 patients showed that people with MS had reduced work and housework productivity even at early disease state and that fatigue is the main problem.
The manuscript is well articulated. The sample is relatively large and representative of the North American population with early/moderate MS. The topic is of great interest.
Despite that, there are some questions.
-No data has been included on the type of work performed, the physical/mental load, and the flexibility or support in the workplace. It is suggested to include this missing data which is of fundamental importance to contextualize the subjects analyzed.
Reply: This is an excellent point and is a limitation to the current questionnaires but certainly in future papers select patients could have a more in-depth narration of their clinical documentation to correlate reasons for fatigue or add further questions into the current questionnaire.
Comments: -Fatigue is reported but not measured with validated scales (e.g. MFIS- Modified Fatigue Impact Scale or FSS-Fatigue Severity Scale), limiting accuracy. Please enter this data. Similarly, in the table are reported MS Symptom Impacted Work, Cognition, Weakness, Pain, but are not reported results. from the scales. Are there data relating to the quantification of pain? of weakness? And relative to cognitive data?
Reply: Thank you for your suggestion. This data was not collected by the registry. For future registry and studies, these scales may be worth built in to further refine these and relation to each other.
Comments: - The percentage of the non-Caucasian population who does it include? enter details.
Reply: Thank you for the question. We hope to have detailed report on this for subsequent papers.
Comments: - Since this is an American study, there are no data on the effect of insurance policies and health coverage on employment and absenteeism. Please clarify this.
Reply: Thank you for your suggestion. This data was not collected by the registry. For future registry and studies, this aspect will be very interesting to look into.
Comments: - Check the values ​​reported in the text and in the tables.
Reply: Thank you and we have checked throughout.
Comments: Finally, this is a complex work, with long tables, it is suggested to create a summary diagram, with the key points, as a graphical abstract.
Reply: Thank you for the suggestion. The registration has wealth of data, which we would like to lay all out. The conclusion from the preliminary analysis is relatively simple, perhaps for future papers with more points, we will summary diagram and graphic abstract.
Round 2
Reviewer 3 Report
Comments and Suggestions for Authors
The authors answered the questions